Insect pollination is important in a smallholder bean farming system

Elisante Filemon 1
Ndakidemi Patrick 1
Arnold Sarah E.J. 1 2
Belmain Steven R. 2
Gurr Geoff M. 3
Darbyshire Iain 4
Xie Gang 5
Stevenson Philip C. p.stevenson@kew.org 2 4
1 Nelson Mandela African Institution of Science and Technology , Arusha , Tanzania
2 Natural Resources Institute, University of Greenwich , Chatham Maritime , Kent , United Kingdom
3 School of Agricultural and Wine Sciences, Charles Sturt University , Orange , Australia
4 Royal Botanic Gardens, Kew , Richmond , Surrey , United Kingdom
5 Quantitative Consulting Unit, Charles Sturt University , Wagga Wagga , NSW , Australia
Sandhu Harpinder
Electronic publication date: 2020 Oct 20
Publication date: 2020
Volume: 8
Electronic Location ID: e10102
Received 2020 Mar 6; Accepted 2020 Sep 14
Copyright: ©2020 Elisante et al.
Copyright year: 2020
Copyright holder: Elisante et al.
License: This is an open access article distributed under the terms of the Creative Commons Attribution License, which permits unrestricted use, distribution, reproduction and adaptation in any medium and for any purpose provided that it is properly attributed. For attribution, the original author(s), title, publication source (PeerJ) and either DOI or URL of the article must be cited.
License URL: https://creativecommons.org/licenses/by/4.0/

Keywords: Pollinators, Phaseolus vulgaris, Smallholders, Crop yield, Ecosystem services, Field margins

Funding: Darwin Initiative (Defra/DfID, UK) 22-012 McKnight Foundation 15-111 This work was funded through a Darwin Initiative (Defra/DfID, UK) grant (22-012) to Philip C. Stevenson and McKnight Foundation grant (15-111) to Geoff M. Gurr. The funders had no role in study design, data collection and analysis, decision to publish, or preparation of the manuscript.

==============================
Background

Many crops are dependent on pollination by insects. Habitat management in agricultural landscapes can support pollinator services and even augment crop production. Common bean (Phaseolus vulgaris L.) is an important legume for the livelihoods of smallholder farmers in many low-income countries, particularly so in East Africa. While this crop is autogamous, it is frequently visited by pollinating insects that could improve yields. However, the value of pollination services to common beans (Kariasii) yield is not known.

Methods

We carried out pollinator-exclusion experiments to determine the contribution of insect pollinators to bean yields. We also carried out a fluorescent-dye experiment to evaluate the role of field margins as refuge for flower-visitors.

Results

Significantly higher yields, based on pods per plant and seeds per pod, were recorded from open-pollinated and hand-pollinated flowers compared to plants from which pollinators had been excluded indicating that flower visitors contribute significantly to bean yields. Similarly, open and hand-pollinated plants recorded the highest mean seed weight. Extrapolation of yield data to field scale indicated a potential increase per hectare from 681 kg in self-pollinated beans to 1,478 kg in open-pollinated beans indicating that flower visitors contributed significantly to crop yield of beans. Our marking study indicated that flower-visiting insects including bees, flies and lepidopterans moved from the field margin flowers into the bean crop. Overall, these results show that insect pollinators are important for optimising bean yields and an important food security consideration on smallholder farms. Field margin vegetation also provides habitat for flower-visiting insects that pollinate beans. Hence, non-crop habitats merit further research focusing on establishing which field margin species are most important and their capacity to support other ecosystem services such as natural pest regulation or even pests.

Introduction

Insect pollination contributes to the production of 75% of crop species (Klein et al., 2007; Potts et al., 2016) and can enhance crop quality and yield even in autogamous crops (Bartomeus et al., 2014; Bishop et al., 2016). An increase in seed and fruit set in these crops has been reported to occur when insects can visit flowers (Pounders, Reed & Pooler, 2006; Roldán & Guerra-Sanz, 2006). As these pollinating insects move between crop flowers, they reduce inbreeding by self-pollination and maximize pollen flow, which improves crop quality and yield (Bartomeus et al., 2014). Yield increases resulting from pollinator visitation can arise through enhanced size, number and weight of seeds/fruits (Bommarco, Marini & Vaissière, 2012; Klatt et al., 2013; Tschoeke et al., 2015).

Anthropogenic activities such as agricultural intensification have resulted in large-scale losses of pollinator abundance and diversity (Klein et al., 2007; Kremen, Williams & Thorp, 2002; Whitehorn et al., 2012) and, consequently, this can impact crop yields (Richards, 2001). Decline in beneficial insects globally are predicted to lead to catastrophic outcomes including pollination deficits, resulting in severe declines in global agricultural production (Giannini et al., 2017). This is exacerbated by increasing demand for pollination services as agriculture has become more pollinator dependent (Aizen et al., 2008). Maximum deposition of pollen in flowering crops (and thus yield) is likely to be achieved when there are high numbers of pollinators visiting flowers and moving between non-crop and crop habitats (Cusser, Neff & Jha, 2016). Consequently, the link between pollinator populations, semi-natural habitats and food security is becoming increasingly apparent.

Non-crop vegetation in agrarian landscapes is important in supporting pollinator communities (Garratt et al., 2017) thus supporting these habitats can mitigate against pollinator declines. Considerable data about pollinator declines and efforts to support them through enhanced habitats has been generated from Europe and North America (Balfour et al., 2018), but there is little equivalent information on threatened African pollinators due to rapid environmental changes (Donaldson et al., 2002; Guenat et al., in press; Kotir, 2011). Climate and land use change have altered the vegetation composition in agrarian landscapes and reduced nesting sites and pollen and nectar resources for pollinators (Ferreira, Boscolo & Viana, 2013; Kearns & Oliveras, 2009) but heterogenous landscapes per se do not necessarily guarantee more pollination services (Samnegård et al., 2016). Conservation strategies require specific information about which insects pollinate crops, enabling targeted and tailored conservation interventions (Garratt et al., 2014).

Common beans (Phaseolus vulgaris L.) are consumed as a primary source of protein by low-income households in many developing countries (Katungi et al., 2009). Common beans provide other fundamental nutritional elements (Brigide et al., 2014) as well as being one of the cheapest dietary protein sources (Hillocks et al., 2006). Interventions in bean production systems are continually required to secure and increase yields. Although many species of beans are autogamous, pollination by insects can nevertheless improve yield and quality (Bartomeus et al., 2014; Ibarra-Perez et al., 1999; Kingha et al., 2012). While many studies have investigated the effects of pollinators on crop yield in fruits and vegetables (Klatt et al., 2013; Tschoeke et al., 2015) relatively few have studied beans with most studies on the role of pollinators being on faba beans (Bartomeus et al., 2014; Nayak et al., 2015). Knowledge about pollinator-dependence of P. vulgaris and their common visitors in East African smallholder farming systems, however, is scarce but can be determined through the use of exclusion experiments (Birkin & Goulson, 2015).

This study has therefore explored the degree of pollinator dependence in beans in a small holder-farming context in East Africa and studied the common flower visitors of P. vulgaris that deliver this ecosystem service along an elevational gradient. Elevation has in previous work been shown to influence pollinator diversity and abundance and may influence the contribution of pollinators to bean yields (Classen et al., 2015; Samnegård et al., 2016). We also applied fluorescent dye to field margin plants in order to evaluate the extent to which flower-visiting insects moved from margin plants into the field, to understand the role of the field margin as a resource for pollinators in this farming system.

Materials & Methods

Study area

This study was conducted in the Moshi Rural District, Kilimanjaro, Tanzania and NM-AIST with field research activities approved by Moshi district council. The sites were located at three elevation zones (henceforth, “low”, “mid” and “high) located between 700 m and 1,800 m above sea level (3.2468–3.3481°S, 37.5044–37.5411°E). In total, 12 sites were selected along the slope of Mt. Kilimanjaro, with 4 at each elevation zone. Farmers on all sites were experienced bean farmers with average farm size of less than 1 ha. All sites were selected based on their management history and to avoid the effects of yield influencing factors such as soil fertility, all experimental sites were managed in the same way.

The natural vegetation in the area varied between elevation zones from more savanna woodlands in the low zone to lower montane forest in the high zone (Ensslin et al., 2015). The area has a bimodal rainfall pattern where the long rains fall between March and May while the short rains fall between October and December (Røhr & Killingtveit, 2003; Zorita & Tilya, 2002). The mean annual rainfall ranges between 600 mm in the low zone to 2,000 mm in the high zone while the mean annual temperature ranges between 23 °C in the low zones to 16 °C in the high zone (Appelhans et al., 2015).

Experimental design

Pollinator-exclusion experiment

To evaluate the effects of different pollination systems on bean yield, a local variety (Kariasii) of common beans (P. vulgaris) was planted in a randomized complete block design. For these exclusion experiments, there was a total of 12 sites, where each zone had four sites. Four experimental sites each of 9 m x 16 m (144 m2) were established at each elevation zone. The bean plants grown in all experimental sites followed standardized common bean spacing (50 cm × 20 cm) (Bucheyeki & Mmbaga, 2013). Weeding was carried out manually with a hand hoe, with care taken to avoid disturbing flower production. The experiment involved three treatments: insect/open-pollination (open), hand-pollination (hand) and self-pollination (self). Each treatment involved four bean plants (n) grown in a block size of 4 m2 and there were four replications per treatment i.e. n = 16 per treatment, n = 48 for 3 treatments within each site, and n = 192 in each zone which made a total of 576 bean plants across the whole study. In the self-pollination treatment, bean plants were individually bagged with polyethylene net (A to Z Textile Ltd., Tanzania, mesh width: 0.4 × 0.7 mm) before the onset of flowering to allow self-pollination (Perrot et al., 2018). The mesh holes were small enough to exclude bean pollinators (medium to large bees) (Kasina et al., 2009) from reaching the plant but large enough to allow airflow and sun radiation and thus minimizing the effects of micro-climate (Bartomeus et al., 2014; Klatt et al., 2013). Netting has been considered a highly effective method for pollinator-exclusion experiments to assess the effects of pollinators on crop yield and no micro-climate effects on bagged flowers/plant has been reported (Birkin & Goulson, 2015; Stein et al., 2017; Suso & del Río, 2014). Based on our daily assessment of the bagged plants, all plants were healthy, with no observed issues associated with moisture, pest damage or fungal development. All bean plants involved in the exclusion experiment were thoroughly examined for any insect (pests or flower visitors) and if present, they were removed before bagging.

In the hand-pollination treatment, we used a technique adopted by local plant breeders where anthers from a donor flower containing matured pollen were rubbed against the stigmas, but unlike in selective breeding processes (Drayner, 1956; Luo et al., 2007), the buds were not emasculated in order to permit maximum pollination to occur. Pollen grains used to pollinate beans in hand-pollination treatment blocks were collected from bean flowers of the same variety grown outside the experimental site. Hand pollinated plants were also enclosed in mesh netting (bagged) after hand-pollination to control for any effect of the netting on yield and inspected every two days. All newly opened bean flowers under this treatment were pollinated. For both self- and hand-pollinated plants, the nets were removed after pod set and when flowers had begun to wither and fall.

The open treatment involved random selection of same number of bean plants, but unlike the other two treatments, each bean plant was tagged and left unbagged to allow visits by insects.

Walked transect

Along with exclusion experiment, we established walking line transects along field margins of the same bean fields to determine the richness and diversity of flower visitors, and their use of non-crop vegetation. In each site, a single line 50 m long transect was established in one of the four field margins. The researcher walked the transect at a slow, consistent pace and all flower visitors observed to interact with flowers of field margin plants within 2 m radius of the researcher were identified and recorded.

Fluorescent dye experiment

Fluorescent dye tracking of flower visitor movements was carried out to determine the extent to which bean flower visitors also interacted with field margin plants. In total, 12 sites in a small-scale bean farming area located along the slope of Mt. Kilimanjaro, were selected for this experiment, with 4 at each elevation. The field sites used for this experiment were the same for the bagging experiments. The non-crop vegetation along field margins comprised native and non-native plant species including herbs, shrubs and scattered trees. Most herbaceous plants and shrubs grow naturally along margins while the tree species may either be growing naturally or have been purposely planted by the farmer/owner to offer benefits including boundary delineation, food or firewood.

Yellow fluorescent pigment (Topline Paint Pty Ltd, Lonsdale SA, Australia, supplied by SprayShop, Dry Creek SA, Australia), was applied at a rate of 1 L/100 L water. An agricultural backpack sprayer (Taizhou Kaifeng Plastic & Steel Co., Ltd, Taizhou, China, supplied by Bajuta International Tanzania Limited, Arusha, Tanzania) was used to spray the dye on to the non-crop vegetation in the field margin. This dye remains on leaf and petal surfaces of plants in the field margin until an insect alights, at which point it rubs off on to the surface of the plant-visiting insect (Schellhorn et al., 2004). The sprayed area was approximately 3 m wide along a 50 m strip and 15 L of solution was sufficient to treat the whole designated area i.e., one margin of the field. The spraying time was between 10:00 and 15:00 h when the temperature was moderate and most insects were actively interacting with flowers (Nielsen et al., 2017) and the activity was carried out during the period when beans were at the 50% flowering stage. The timing was chosen to ensure there was maximum potential for interaction between flower visitors and the crop when measuring their use of the field margin.

Data collection

Effects of different pollination systems in common bean yield

Beans from each treatment site were harvested after reaching senescence and the mean number of pods per plant, seeds per pod and weight of 30 representative dry seeds were calculated to determine the treatment effect. All three response variables (number of pods per plant, seeds per pod and weight of seeds) were tested for correlation using R software. Also, the average yield data were converted according to typical planting density and used to calculate bean yield (kg ha−1). Initially, the average mass of seed was calculated from 30 representative seeds per plant. The average number of pods per plant for each treatment were obtained from four (4) bean plants of each block. It was then multiplied by 200,000 plants to get the average number of pods per hectare, multiplied it by average number of seeds per pod. The average bean yield (g ha−1) were then obtained by multiplying the average number of seeds ha−1 with the average weight per 30 seeds (g)/30, multiplied it by 1000 to get the average bean yield (kg ha−1). To obtain the average income, we visited three local markets in the study area and the average price of beans was around 1518 Tanzanian shillings per kg. This value was then used to calculate the differences in average income generation per hectare if beans harvested from each treatment site would have been sold in local markets (Table 1).

Table 1 Mean bean yield from three pollination treatments (open, hand and self) Ha−1, percentage increase on self-pollinated plants, mean dividend (1518 TSH per kg) from three local markets in the study area converted to USD currency.

The exchange rate was 1USD to 2200.00 Tanzanian shillings (obtained from CRDB Bank Plc., Arusha).

Pollination treatments	Average bean yield (Kg Ha−1)	% Increase in bean yield	Average IncomeHa−1(USD)	
Open	1,478	117	1,020	
Hand	1,131	66	780	
Self	681	–	470	

In the field margins, any insect that interacted with a flower within a line transect was recorded. A visit was defined to have occurred when the visitor’s body came into contact with reproductive organs of the flower (Lundgren, Lázaro & Totland, 2013). The insect counts were done during the flowering period at the same time as the exclusion experiment was being conducted. Unidentified specimens were collected using a sweep net, and preserved in 70% ethanol for subsequent identification in the laboratory. The recorded numbers of insects were then used to calculate the abundance and diversity for each flower visitor across three elevation zones.

Effect of field margin vegetation to pollinator numbers in bean field

Insects were sampled from the crop using sweep-nets 24 h after spraying margins with fluorescent dye and repeated for three consecutive days. Samples were taken at four distances from the edge bordering the sprayed field margin i.e., 0 m, 10 m, 20 m, and 40 m (Perović et al., 2011). At each distance, the sampling transects, 50 m long and 3 m wide, ran in parallel with the control transect (i.e., field-margin edge, 0 m). They were surveyed using sweep nets between 10.00 and 15:00 hrs. Insects were sampled when the weather was sunny with moderate ambient temperature of above 22 °C to avoid the effects of low temperature which reduce foraging activity of most insects (Mellanby, 1939). The collected samples were killed on site with ethanol-soaked tissue in a vial, kept in a -20 °C freezer and later sorted for identification in the lab. Each insect sample was inspected for pigment under UV-light. The insect was considered marked (to have pigment) when a clear drop pattern of the dye was observed on any part of the body while samples found only to have small, scattered stains were regarded as unmarked and were considered contaminated during sampling in sweep net (Schellhorn et al., 2004).

Statistical analysis

There was a significant correlation between dependent variables: number of pods per plant, number of seeds per pod and weight of seeds. Because the variables correlated significantly with each other, a multivariate analysis of variance (MANOVA) was then performed to determine the overall effects of pollination systems on bean yields across the zones. A full factorial model was fitted and combined four potential predictor variables: treatment, zone, sites and season. The means and standard errors of means between treatments on each dependent variable were then estimated based on the univariate ANOVA models obtained from optimal MANOVA model. A univariate ANOVA was also used to determine the effects of field margin position on numbers of flower visitors in the bean field. Tukey’s honest significant difference (HSD) test was then applied for multiple comparisons of means at 95% - confidence level to understand where those differences lay between the treatments. A Kruskal–Wallis rank sum test (KW) was used to determine the significant differences between the proportions of dye-marked versus unmarked insects by zone and sampling days. The Shannon Diversity Index (H′) was used to determine insect functional group diversity across elevation zones (Shannon, 1948): H′=−∑i=1kpilnpi

Where: H′= the Shannon diversity index; pi = proportion of each species in the sample; ln(pi) = natural logarithm of this proportion.

In this study, some data were analyzed using R version 3.4.0 (R Core Team, 2017) and some were analyzed using STATISTICA 8.0 version 7.

Results

Effects of pollination service on yield components

All three responsible variables (number of pods per plant, seeds per pod and weight of seeds) which were tested showed significant positive correlation to each other. Open-pollinated plants to which flower visiting by insects was permitted bore the highest number of pods, had the highest mean number of seeds per pod, and the mean weight of individual seeds was also highest, compared to the self-pollinated plants from which pollinating insects were excluded (pods: F = 166.5, df = 1, p < 0.001; seeds: F = 101.9, df = 1, p < 0.001; weight: F = 38.08, df = 1, p < 0.001). Yields of pods and numbers of seeds per pod in hand-pollinated beans did not differ significantly from the open- pollinated (unbagged) although individual weight of seeds was lower, possibly reflecting a minor effect of method (Fig. 1). Also, the Tukey’s honest significant difference (HSD) test showed significant differences between hand and self-pollinated plants (pods: p < 0.001; seeds: p < 0.001; weight: p < 0.001). The highest pod count, bean/pod count and seed weight overall was consistently recorded from the open-pollinated (unbagged) plants in the mid-zone. Although we found significant differences among zones (F = 26.604, df = 2, p < 0.001), there were no significant differences between treatments and the zones (F = 0.565, df = 4, p = 0.8709).

Figure 1 Bean-yield parameters, mean (±SE) number of pods (A), number of seeds (B) and weight of 30 seeds (C) for each treatment.

The treatments are: open-pollination (open), hand-pollination (hand) and self-pollination (self). The error bars on top of the means measure the Least Significant Difference (LSD). Pollination treatments are considered significantly different if the error bars do not overlap, (F = 36.96, df = 2, p < 0.001).

We found significant differences in the abundance of insects over three elevations (KW = 7.2728, df = 2, p = 0.0264) where the mid zone recorded the highest abundance of insects (430) compared to the low zone (390) and the high zone (107). The results also showed that the abundance of collected insects during the short and long rain seasons did not vary significantly (KW = 2.9477, df = 1, p = 0.086). Insect species diversity in the low zone (H′= 3.0742), mid zone (H′= 3.0809) and the high zone (H′= 3.0693) were almost identical to each other. However, honeybees (Hymenoptera: Apidae: Apis mellifera) were the most abundant functional group in the mid zone (33% of the total) followed by small bees (Hymenoptera: Halictidae and Apidae) (10.2%). Similarly, we recorded a high proportion of honeybees (24.3% of the total) within the total catch from the high zone, followed by small bees (18.2%). Unlike the mid and high zones, the most abundant group in the low zone was small bees (23.3% of the total) then followed by honeybees (21.5%). Other recorded flower visitors that were common across all three zones were butterflies and moths (Lepidoptera), hoverflies (Diptera: Syrphidae), beeflies (Diptera: Bombyliidae), wasps (Hymenoptera), carpenter bees (Hymenoptera: Apidae: Xylocopa spp.), flower beetles (Coleoptera) and ants (Hymenoptera). Amegilla bees (Hymenoptera: Apidae: Amegilla sp.) and solitary bees (Hymenoptera: Apoidea) were recorded at small proportions across the zones.

The potential value of insect pollination in bean yield and income generation

When we extrapolated the bean yields per plant to field level based on typical planting densities, the increase in kg ha−1 as a result of insect flower visits became clear (Table 1). There was an increase in mean yield per hectare from 681 kg in self-pollinated beans to 1131 kg and 1478 kg in hand-pollinated beans and open-pollinated beans respectively. Variability in these estimates is illustrated in Fig. 1. from which they were derived. Due to increased bean yields following insect pollination, the calculated average income per hectare was highest in open-pollinated bean blocks compared with the other treatments (Table 1).

Movement of pollinators between field margins and bean field

A total of 980 insects were sampled of which 327 were flower-visiting taxa that may be pollinators (Corlett, 2004; Larson, Kevan & Inouye, 2001). Pollinators were observed under UV light and a total number of 203 (62%) insects tested positively (dye-marked) and 124 (38%) insects tested negatively (unmarked). However, the number of dye-marked (KW = 2.926, df = 2, p = 0.2315) and total sampled (KW = 1.792, df = 2, p = 0.4082) insects did not vary significantly between the zones. Bees overall were the most abundant marked taxon (Fig. 2) with honeybees the most frequently sampled dye-marked species across the zones. A total of 103 (51% of the total insect catch) honeybee individuals were collected during three days of sampling. Overall, honeybees were the most often sampled species while cuckoo wasps (Hymenoptera: Chrysididae) were the least sampled species during this assessment. Other sampled flower visitors included Amegilla bees, beeflies, hoverflies, butterflies, moths and a diversity of small solitary bees. The number of dye-marked insects did not vary significantly between sampling days (KW = 3.963, df = 2, p = 0.1379). However, the number of marked insects caught varied significantly by distance from the margin (F = 8.3127, df = 3, p < 0.0001) with most marked individuals being sampled nearer to field margins (Fig. 3). It was also found that the abundance of dye-marked insects such as honeybees did not decline with distance; 0 m (50%), 10 m (13%), 30 m (21%) and 40 m (16%) while insects such as hoverflies, small bees and butterflies declined with increasing distance from field margin.

Figure 2 The proportion of dye-marked insects by functional group collected during fluorescent-dye experiment in northern Tanzania.

Figure 3 The effects of field margin position on numbers of flower visitors in bean field.

The field margin here is indicated as 0 m. The error bars on top of the means measure the Least Significant Difference, and different letters within the same group (distance) shows significant differences (p ≤ 0.05).

Discussion

It is often assumed that common beans are largely autogamous and that, consequently, the role of pollinators is trivial (Ibarra-Perez, Ehdaie & Waines, 1997; Papa & Gepts, 2003). Here we show that pollination can make a substantial, and financially significant contribution to yield. Indeed, our calculations indicated that the value of insect pollination was relatively high and farmer could face a potential loss of up to $500 of their income per hectare if insect pollination services were lost. This loss could be greater still where farmers can harvest two crops per year. In a country where the Gross National Income per capita in 2017 was below $1000 (World Bank, 2018) for a farm of around 1 ha in size this is a major loss to household income and food and nutritional security, thus pollination services and landscape management to conserve pollinating insects should be a major consideration in drafting agricultural policy to enhance food and nutritional security in bean farming systems. By increasing insect pollination services in this agri-system, farmers have the opportunity chance to improve yield of other bean varieties such as Uyole 90, Uyole njano, Rose coco, Kijivu local variety, Jesca as well as other non-bean crops and fruits which are commonly grown in the area. The study suggests that sustainable crop yield is possible among smallholder farmers in the study area by maximising pollination services, and conversely that income losses can be avoided by farming practices that reduce risk to pollinator populations, such as excessive spraying of pesticides. However, more information is needed on which species are the most important pollinator of bean crop and which specific field margin plants are more important in supporting them.

Open pollination increased bean yield and quality through seed weight, seed number per pod, and pod number per plant. Increase in weight in unbagged beans is an indication of improved seed yield brought about by pollinating insects (Douka et al., 2018; Ibarra-Perez et al., 1999). We recorded no trade-offs related to open pollination with respect to yield. The result concurs with other studies such as Kingha et al. (2012) who recorded high yield benefits from unbagged common beans but contrast with the study by Free (1966), who reported only moderate yield benefits of unbagged common beans visited by honeybees. The role of honeybees versus wild bees is likely to be key to understanding which flower visiting species are important to yield in these cases: increasing evidence indicates that honeybees are not always the most efficient or effective pollinators (Garibaldi et al., 2013; Grass et al., 2018), including in legume crops where they are among the most frequent flower visitors (Marzinzig et al., 2018). Honeybees (51%) were the most frequently sampled insects and particularly in the mid and high zones. This could have been contributed by bee-keeping activities but also most farms in this area comprise diverse trees, shrubs and herbs providing potential forage for honeybees (Fernandes, Oktingati & Maghembe, 1985). Other comparable studies in other parts of East Africa have also reported A. mellifera as the most abundant flower visitor in cropping systems (Kasina et al., 2009; Otieno et al., 2011). Other flower visiting insects collected were Amegilla sp. (2%), beeflies (2%), carpenter bees (3%), hoverflies (6%) and miscellaneous Lepidoptera (13%), all of which could play a role in pollination. Other work on pollination in common beans has indicated that short-tongued bees rob heavily, whereas long-tongued species are effective pollinators (Kingha et al., 2012; Ramos et al., 2018). Although apparent evidence of robbery as indicated by holes chewed into corollas is not necessarily indicative of a major impact on fertilization, robbery events are typically much less frequent than pollinating visits (Barlow et al., 2017). In East Africa, long-tongued bumblebees (Bombus spp.) are not present but carpenter bees fill a similar niche and while highly effective as bean pollinators (Masiga et al., 2014) are also nectar robbers (Irwin et al., 2010). The presence of carpenter bees in bean fields could have increased visitation of honeybees to common bean flowers; where this was associated with nectar robbery by the Xylocopa spp. this could provide foraging opportunities for honeybees, which are secondary robbers. However, further investigation is required to determine whether this influenced bean yields through increased flower visitation. Honeybee visitation might also increase as a result of heterospecific social learning, in which carpenter bee visits are observed and used to identify a safe and prolific nectar source by honeybees (Leadbeater & Chittka, 2007). We would recommend further work in our system to investigate the efficacy of pollination services offered by specific flower visitors and those that interacted with common beans during sampling.

We would recommend further work in our system to investigate the efficacy of pollination services offered by specific flower visitors and those that interacted with common beans during sampling.

Our exclusion experiments demonstrated that open-plants yielded more than self-plants. Low yield in self-plants was likely due to the lack of visitation by insects and transfer of pollen between plants after excluding flower visitors which might have lowered both pods and seed production (Ibarra-Perez et al., 1999) as opposed to hand- plants which received pollen after being pollinated manually. Another explanation could be that common bean flowers do not activate well without insect visits therefore fewer pollen grains contact stigmas of self-pollinated flowers for fertilization. As the insects forage, they move/shake flowers which increases pollen-stigma contact and augment fertilization (Mainkete et al., 2019). Yield from hand-plants did not differ significantly from open-plants with respect to pods per plant and beans per pod although the mean weights of individual beans were slightly lower. This may be a minor effect of bagging the hand-pollinated plants or that the experimentally applied single pollination event was insufficient to optimise yield and this may have affected fruit setting among plants (Otieno et al., 2011). More typical is to leave the plants in hand-pollination treatments uncovered (Birkin & Goulson, 2015; Grass et al., 2018) although this may then not control for the effect of the bag on photosynthesis and metabolism. While this means it was therefore not possible to evaluate completely whether this agricultural system was pollinator-limited, it did provide important information about the contribution of pollination in this crop, specifically that allowing insect visitation to flowers dramatically increases yield in this otherwise autogamous crop, and therefore if pollinator numbers are low yield may be limited. Therefore, determining pollination services should be a major priority in policy-setting in bean farming, as our results have demonstrated that insect pollination provides a major contribution to yields and is an essential ecosystem service in supporting food security in bean agri-systems.

Based on the finding that pollination is important and valuable, we also evaluated whether potential pollinators in the crop were making use of natural and semi-natural vegetation around field margins, as this is a key target for management interventions to promote pollinator species (Potts et al., 2016). Capturing various dye-marked insects from within the crop is therefore evidence that the insect has previously visited the margin either for feed or refuge before moving into the crop. Although we also found other non-pollinating species, including pests, during collection, they were not analysed specifically since our target was pollinating insects. As our study shows evidence of frequent movement by flower-visiting insects from the margin to the crop, indicating a role of the margin in providing resources for these insects. However, further studies should explore whether these insects are using field margin vegetation as a resting, nesting, food resource sites or both. In the case of potential pollinators, this can be associated with feeding behaviours in both the margin and crop.

A high proportion of the insects collected from the crop contained dye traces, which indicates extensive movement between crop margin and crop in a distant-dependent fashion, with more margin-users found very close to the margin. This demonstrates that firstly, not all margin insects remain in the margin in this system, so the margin can be a donor of ecosystem services into the crop. Secondly, penetration of these services into the crop has the potential to reach the centre of the field but will be most marked around the edges, close to the margin unless alternative management techniques such as intercropping or sowing of flower strips within the field are used to enhance movement around the fields (Korpela et al., 2013; Pereira et al., 2015). However, there was no significant difference between the proportions of marked potential-pollinators at 10, 20, 30, or 40 m, implying two behavioural syndromes among margin-users in the crop, those that strayed only a short distance (0 m) into the crop, or those who moved off margins and into the crop and then foraged more widely among the crop plants. For instance, dye-marked insects such as honeybees were sampled at all distances. The total number of dye-marked honeybees captured at each distance were 50% (0 m), 13% (10 m), 21% (30 m) and 16% (40 m), suggesting that honeybees can forage up to over 40 m and there was no evidence of distance-dependent effect recorded for this insect over 10 m. Similarly, Woodcock et al. (2016) reported no declining effect in honeybees’ visitation rates into the oilseed rape field even at a distance of 200 m from the field edge.

Surprisingly, we did not sample marked beeflies at any distance in the bean field and instead all marked individuals were collected at field margin (0 m). The explanation could be that beeflies are not able to effectively feed from common beans and so seldom have reason to enter the crops or fly a large distance into the field to forage. As the fields were small, it was unsurprising that more robust flying insects (that can cover moderate distances of 100 m or more in a short time) dominated samples from the centre of the field. This is particularly the case for carpenter bees (Pasquet et al., 2008) and honeybees (Beekman & Ratnieks, 2000), which used the majority of the field fairly evenly. This contrasts to work on coffee plantations that are very large, in which there are strong distance-dependent effects moving away from semi-natural habitat at the edges of fields, but again this is especially observed for small bees (Klein, Steffan-Dewenter & Tscharntke, 2003). Similarly, in large fields of temperate oilseed rape, the number of bees towards the field centre can be very low (Bailey et al., 2014). We suggest that future studies should also consider the effect of field size and landscape patterns on the abundance and richness of pollinators in smallholders’ bean fields. However, it is important to note that this study did not focus on monitoring absolute abundances of potential pollinators at different distances, but on the eventual destinations of field margin users, and the sweep netting technique did not discriminate pollinators from nectar thieves or transient insects not using the flowers.

However, as nearly 50% of potential pollinating species sampled even from the centre of the field showed fluorescent dye marks consistent with use of the margins, our study highlights that the margin vegetation is providing benefits to these insects. Plant species such as Ageratum conyzoides, Commelina foliacea, Desmodium intortum, Morus australis and Tithonia diversifolia were commonly sampled in the field margins of the study site (Elisante et al., 2019). This study also revealed a high diversity of insects across all three zones suggesting that pollination service necessary for bean yield may not be limited in bean agri-system due to a high abundance and diversity pollinating insects. As in the fluorescent dye experiment, bees were the most dominant taxa along field margins of bean fields. Our flower visit observations and other studies (Kasina et al., 2009) indicate that they are major pollinators of both cultivated crops and wild plants in this agri-system. For farmers, the high use of field margin plants by bees also associated with crop demonstrates that field margin plants may be important in maintaining potential pollinators of bean crop in the bean field. Since the measurement from fluorescent dye experiment represents the maximum potential interactions between flower visitors and common beans, this may be enhanced and supported through proper management of field-margin vegetation adjacent to the crop field. Other studies have also reported that presence of diverse and floral rich margins can enhance pollinator species in the neighbouring crop field (Garratt et al., 2017; Morandin & Kremen, 2013). However, further work should focus on characterising the nature of insect-plant interactions in the margin and crop to indicate which plants are most important for promoting pollinator abundance and movement into the crop. This study suggests further studies also to focus on comparing how different types and management of field margins can affect stability and persistence of pollination services in this agri-system.

Conclusions

This study aimed to establish the contribution of flower visiting insects to yield in bean crops. We revealed that insect pollination offers a significant benefit to yield in common beans in East African smallholder bean agri-systems. Following this evidence, we argue that biotic pollination is as important as other agricultural inputs to improve crop productivity and nutritional and food security since it provided a yield boost of 117% relative to beans from which insects were excluded. This is similar to (or exceeds) the impact of many recent interventions reported in agriculture in low-income systems (Koskey et al., 2017; Pretty et al., 2006). However, farmers need to understand such services as necessary for them to maximise yields and recognize the importance of managing agricultural biodiversity in their farmlands. This is currently a limiting factor as many farmers are knowledge-poor about beneficial invertebrates (Elisante et al., 2019).

We found a high proportion of pollinating insects captured in the crop had previously visited the margin, suggesting that field margin plants can act as refuge or food reserve for important pollinators. This use of margins indicates the need for sustainable management interventions that protect natural vegetation, in order to augment pollinator abundance and pollination services in agrarian landscapes (Boreux et al., 2013). During the off-season and when beans are not blooming, these plants can support pollinators by providing food and nesting sites and thus keeping their numbers at natural state (Morrison et al., 2017). We argue that farming practices that threaten agricultural biodiversity in bean farming systems, such as removal or burning of field margins, should be discouraged and instead, farmers will see benefits if empowered to practice ecological-intensification (Potts et al., 2016). Our study was confined to only one local variety of common beans; future studies can expand and explore how production of different bean cultivars respond to pollination by insects. Cultivars of common beans differ in flowering time but may also attract different groups of pollinators based on flower morphology but also the quantity and quality of nectar they produce. Further studies on pollination ecology of common beans may also need to look at two important aspects; pollinator-specificity and effectiveness, to determine which insect species is the most effective pollinator of this crop.

We thank farmers who allowed us to conduct our experiments in their bean fields during this study.

Additional Information and Declarations

Competing Interests

Author Contributions

Field Study Permissions

Data Availability

The authors declare there are no competing interests.

Filemon Elisante and Sarah E.J. Arnold conceived and designed the experiments, performed the experiments, analyzed the data, prepared figures and/or tables, authored or reviewed drafts of the paper, and approved the final draft.

Patrick Ndakidemi and Philip C. Stevenson conceived and designed the experiments, prepared figures and/or tables, authored or reviewed drafts of the paper, and approved the final draft.

Steven R. Belmain conceived and designed the experiments, analyzed the data, authored or reviewed drafts of the paper, and approved the final draft.

Geoff M. Gurr and Iain Darbyshire conceived and designed the experiments, authored or reviewed drafts of the paper, and approved the final draft.

Gang Xie analyzed the data, authored or reviewed drafts of the paper, and approved the final draft.

The following information was supplied relating to field study approvals (i.e. approving body and any reference numbers):

Field Experiments were approved through Nelson Mandela African Institution of Science and Technology by Moshi District Council, Tanzania.

The following information was supplied regarding data availability:

Raw data available at Open Science Framework: Arnold, S. (2020, September 18). Raw data sets Pollination and yield manuscript. Retrieved from https://osf.io/3uv68.

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
