# Peer review of "Insect pollination is important in a smallholder bean farming system"

_PeerJ, doi:10.7717/peerj.10102_

## Round 0.1 · original submission · Major Revisions

There are a number of comments that may be dealt in minor revision of the paper. However, Reviewer 2 noted some gaps in literature on P vulgaris pollination and pollinators that are not covered sufficiently in the manuscript. In addition please attend to the comments to improve discussion as well.

Reviewer 1 ·

Basic reporting

The paper is clear, well written in professional English and structured according to PeerJ standards. Appropriate literature had been cited with no major omissions and no superfluous or inappropriate citations. The figures are relevant and of good quality although I have some comments on the labels (see general comments)

The research addresses differences in yield of Phaseolus vulgaris L (common bean) by employing three treatments: open pollination, hand pollination and self-pollination, using standard methods that have been widely published. This experimental approach was augmented with transects that recorded flower visiting insect presence together with a mark and recapture study that investigated the extent to which pollinating insects observed in the crop were using local semi-natural habitats in field margins. Combining these results the authors drew the following conclusions: insect pollination increases been yield; that this is important because it has a beneficial economic impact for small-holder farmers; that insects that pollinate the crops also use the margins, that margins could thereby be important in supporting pollination in the bean crop studied; that although the authors found no pollinator limitation, the potential for limitation is a risk for small-holder farmers and that, by extension, the provision of non-crop habitat could increase security for farmers. The authors carried out the study along an altitudinal gradient, at three different altitudes and found the treatment effects consistent across altitude. This topic fits within the scope of PeerJ being a Research Article in the field of Environmental Science. The research gap was clearly defined.

Experimental design

The experiment was well designed and described. The results were credible. The authors carefully and clearly linked the results to the original research question and the implications of the findings clearly are described in the context of the evidence base.

The field study permit is in order.

Validity of the findings

As I said above I can’t download the data so I am afraid I will have to ask the Academic editor to check it unless there is another way for me to access it.

The authors have acknowledged the gaps that are in their research and presented negative results where they found them. The replication is sufficient (see my notes in general comments).

Additional comments

Overall I really liked (and enjoyed) this paper, apart from the minor points below, it is very clear and delivers fundamental information that's useful, in a concise and readable article.

I have some small notes that I would like to share with the authors:

Line 34 – It would be useful to have the yield linked to area unit (ha) in the abstract as that makes it immediately clear for the reader. It might also be good to see the estimated economic benefit specified but that's not essential.

Line 111 – It surprises me that there is no difference in soil fertility with altitude, especially in the light of different natural vegetation. How was this determined, were there soil tests? Could you add this, or say how you decided that this was so?

Lines 124 -132 I find the replication a little confusing. As I understand it you have:

3 x altitudes
4 x blocks (you call them plots of 9m x 16m, maybe they should be called blocks)
Within each block each treatment was applied once – in plots of 4m2 with 4 beans each
So you have 4 x 4 = 16 bean plants per treatment at each elevation and a total n of 48.
Is that correct? It would be really helpful to have n stated.
Later (Line 157) you say 12 sites for the three elevation zones –maybe this could be brought up earlier and the experimental design slightly clarified, I think you have 4 replicates of each ‘treatment’ at each elevation - each replicate based on four plants which presumably you averaged for analysis. Sorry to labour this point!

Line 134 – you do not state was ‘raw’ means – it might not be clear for the lay reader as ‘raw’ self-pollination is not mentioned again –being referred to thereafter just as self-pollination.

Line 166 - Did you identify any of the plants in the margins? It would be interesting for the reader to know something about the composition of the margins if you have it.

Line 194 - It would be helpful to have the information about when the net bags were removed earlier in the methods – this was a question that I queried as I was reading.

Line 204 – insert an ‘a’ in the first line so that it reads ‘ …any insect that interacted with a flower …’

Line 228 – I think here it would be better to say that you tested for correlation (and how) in the methods, and then in results give the results of the correlation (plus statistics). The details of the MANOVA and ANOVA are a little sketchy, stating the model you used would be useful (you say across the zones, how was altitude included in the analysis?) and how was field margin included in the ANOVA? Were any of the data transformed before analysis? Did you test for normality? These details would be really useful and would be reassuring.

Line 266 - With reference to the presence of Apis mellifera and the increase in abundance in the mid-zone – presumably this is due to more beekeeping going on in this area, if you have any information to shed light on this, it would add context.

Line 279 – replace ‘particularly apparent’ with ‘clear’
Line 297 – remove ‘at all’

Discussion: This lightly raises the issue of nectar robbing, I don’t want to open a large can of worms here for the authors but if you see any potential to address this a little more deeply it would be interesting. The authors found that A. mellifera marked with dye in the margin were found in the crop, yet A. mellifera would not be expected to a good pollinator in beans because they are short-tongued – in the UK we have observed that honey bees do visit beans, but that they usually use holes cut by Bombus species to rob. This isn’t a big issue, but it raises a question about how meaningful the incidence of honey bees is in this paper.

Comments on Figure labels
Fig. 1. Bean yield parameters, mean (±SE) number of pods, number of seeds and weight of 30
seeds for each treatment.

The treatments are: open-pollination (open), hand-pollination (hand) and self-pollination
(self). The error bars on top of the means measure the Least Significant Difference (LSD).
Pollination treatments are considered significantly different if the error bars do not overlap (p≤ 0.05).

This is a little misleading, the significance was determined by a statistical test – the use of error bars demonstrates the differences, but this is not how the statistical differences were determined (as suggested by the label) – for completeness the statistics (n, F, p, df) could be added, either as an annotation in the figure, or in the figure label.

Figure 2
The proportion of dye-marked insects by functional group collected during fluorescent dye
experiment in bean agri-systems.

The location of the experiment could be stated, rather than the generic ‘bean agri-systems’ that is used.

Figure 3
The effects of field margin position on numbers of flower visitors in bean field. The field
margin here is indicated as 0 m.

The label does not explain what the small letters indicate, this should be done and the statistics could be included so that the figure can be interpreted without reading the text.

Reviewer 2 ·

Basic reporting

Text is generally clear and unambiguous. See line-by-line comments for some areas where clarity could be improved. A main point you need to be clearer on is how you moved from the yield parameters that you measured (pod per plant, seed per pod, mass of 30 seeds) to yield in kg/ha.

I think you need to include more references specifically related to Phaseolous vulgaris pollination - you were perhaps overly reliant on referring to the literature on Vicia faba which is a different species with a different floral structure. From a brief literature search, I found quite a lot of literature on P vulgaris pollination and pollinators, there is scope to include a lot of that literature here.

The figures and tables should be clearer about what error bars are presented. I suggest you also need to include some measure of variability in table 1. It would aid transparency to include your raw data (or a link to where this can be accessed). You should include more mention of effect sizes in the results text.

Experimental design

Research questions are well defined and meaningful. The investigation appears to have been rigorous and conducted to a high standard. The methods are generally described with enough detail for them to be replicated, see the line-by-line comments for areas where more details or clarification is required.

Validity of the findings

The data on which the conclusions are based don't seem to have been made available in a repository.

There are some parts of the discussion section that need to be improved and are a bit tenuous. I specify these in the line-by-line comments. Generally I think more should be made of the comparison between bagged plant treatments (hand pollinated or selfed) as this is the properly controlled comparison.

Additional comments

This was a well thought out study that helps to address knowledge gaps around the value of pollination services in East Africa and associated agronomic management in smallerholder farming systems. It is an interesting and generally well written paper. Below I list suggested changes required to make the article suitable for publication.

Line by line comments
First page – ‘field margins’ not ‘field margin’. The third sentence in your methods section is describing results.
25 – specify this is common bean. I think this statement is incorrect and there is already evidence about the effect of insect pollination on Phaseolous vulgaris yield. For example, see https://doi.org/10.1371/journal.pone.0204460, https://doi.org/10.1080/00218839.1966.11100139, https://doi.org/10.32861/10.32861/jac.411.105.111
27 – change ‘margin’ to ‘margins’. Have you actually tested the importance of margins as a refuge, or have you tested whether margins increase provision of pollination service?
28-31 – this text is results not methods and should be moved accordingly – or delete and replace with text explaining your different treatments and introducing what the marking study involved.
32-35 - I think it would be valuable to mention your hand pollination treatment here too as this is the properly controlled comparison – your difference between open pollinated and excluded plants may be confounded with an effect of bagging.
52-54 – insect pollinators can also stimulate self pollination
58-59 – reference needed for this first point that agriculture intensification has resulted in large scale losses of poll abundance and diversity. Perhaps reflect that this is a complicated process and several factors have contributed to pollinator losses.
71 – be clearer what you mean by ‘and their support’ – it is support of pollinators, not of pollinator declines?
74-75 – it’d be more effective to include a ref about changes in agri environment in Africa here to link to your previous sentence
91 – I suggest autogamous should be removed here. Faba bean is partially autogamous.
93 – see comment above about some different studies that have tested P vulgaris pollinator dependence in a range of environments
109 – clarify how many sites there were in total and how many per elevation zone (e.g. like on lines 170-172)
124 – how widespread is Kariasii, approx. what percentage of growers use it? (how representative is this variety of beans in the surrounding landscape) Plant genotype can be very important in terms of reliance on pollination.
125-126 – were these experimental plots at one site, or distributed across different farmer’s land within each elevation zone?
139 – I think it is an oversimplification to say no micro-climate effects have been reported. Free 1993 (Insect pollination of crops book) discusses this in some detail. See doi:10.1093/jxb/erw430 table 2 for a test of bagging effects. Alternatively, your bagging treatment could also be excluding important pest species.
160-166 – Rewrite this to clarify that the transect was from the margin to 50m into the field, not just along the field margin.
170-172 – were these the same sites as the exclusion experiment?
185 – change ‘during which’ to ‘and’
188-190 – You could reflect in the discussion that this measurement therefore represents the maximum potential interaction between flower visitors and the crop.
197-198 – how did you convert from yield data to kg/ha? So you measured weight of 30 representative dry seeds, not the total seed mass per plant or per unit area. If the pollinated plants produced a greater number of seeds in total, and those seeds were heavier, did you account for both of these factors in your yield conversion?
242-243 – change wording, were all stats analyses done in both statistical software?
245-257 – in the results section it would be very useful to state effect sizes – how big were the differences between these pollination treatments? What were the significant differences among zones?
251-253 – perhaps the more important comparison here is between the hand-pollination and self-pollination treatments as these control for the effect of bagging, the difference between hand pollination and open is less interesting in the context of pollination dependence due to confounding effects of the bags.
280 – include comparison hand-pollination vs self-pollination here
282-284 – this is discussion text. This could indicate that there is no pollination deficit, but could alternatively indicate that your hand pollination treatment is not as effective as insects, or that bagging negatively impacted on plant growth
297 – should this be ‘of all’ not ‘at all’?
303 – additional description of the results here would be useful – how far in to the field did the insects travel? You could move some of the description (presentation of results) from the discussion section (e.g. lines 390-400) to here.
310-311 – this is an interesting discussion point – it would useful to clarify here how many bean crops the farmers can typically produce per year
324 – suggest you remove mention of quality if not measured here. Increase in total weight does not indicate seed quality – there could just be more seeds (and these could even be smaller)
325-328 – P vulgaris is a different species to faba bean, I think you should try to avoid so many comparisons to faba bean and instead include more literature about common bean pollination dependence. What are the modest benefits that Free 1966 recorded? See some refs as examples in the comment above.
332 – if honeybees are the most frequently sampled insects, how does this change your management recommendations?
337-341– Marzinzig et al 2018 relates to faba bean which has a different floral structure – switch to a reference about nectar robbing of P vulgaris flowers. Regarding the second part of this sentence, frequency of robbing vs pollination visits, the Marzinzig et al 2018 study that you cite in the same sentence found 47.4% of visits were robbing compared to only 28.6% pollination.
348 – I don’t agree with this - inbreeding depression would likely act on the offspring (and yield in subsequent generations if those seeds were planted) rather than current generation productivity. If you want to expand on this argument I think you need to include some refs specifically about retention of selfed vs outcrossed pods (there is some limited evidence in faba bean). This looks like it might be an interesting ref for P vulgaris - DOI: 10.2307/2444124
350-352 – include specific refs regarding P vulgaris rather than referring to leguminous flowers in general
353 – you say that pod per plant and beans per pod were not affected between hand pollination and open, but seed mass was, this suggests that fertilization was adequate in the hand pollination treatment but that the bagging reduced plant resource availability and ability to mature the seeds compared to open plants. (looking now at figure 1, it looks like in fact the difference between hand and open across these different yield metrics is quite similar).
363-368 – this viability of hand pollination by growers discussion is quite a stretch and should be removed – you just discussed on line 354 that differences may be due to bagging effect and not the pollination event itself
373-384 – you didn’t measure non-pollinating species / pest species but it would be useful to include a ref about it here, whether this vegetation in field margins can increase pest density in the crop
398 – clarify what percentages refer to here, I assume it is % of total insect number measured across all distances?
416 – doesn’t this contradict the Woodcock et al 2016 study you cite on line 401 also regarding oilseed rape? How do these bits of information fit together?
464-466 – good to include this, cultivar is likely to be highly important. You could change cross-pollination to ‘pollination’ – insect visits can increase both self and cross pollination.
Table 1 – For transparency you should include standard deviation or standard error in the yield measurement, and then show how this translates to uncertainty in your % yield increase and economic estimate.
Figure 1 – if you did the Tukey HSD tests it would be clearer to show what was significantly different by using letters above the bars to distinguish the significantly different groups – like you do in figure 3. Clarify in figure legend whether error bars show SE or LSD.
Figure 3 – explain what the error bars show there.

---

## Round 0.2 · Minor Revisions

We now have received second round of reviews. One reviewer agrees with all the revisions. However, there are some minor changes that are suggested by second reviewer. I hope you can address them before we make a final decision.

Reviewer 1 ·

Basic reporting

This is a second review after corrections. The majority of my comments stand, therefore I am only commenting on the changes that have been made.

Experimental design

No comment.

Validity of the findings

No comment.

Additional comments

Thank you for addressing the minor comments in my previous review. The manuscript is now ready for publication and this paper makes a valuable contribution to our understanding of pollination in smallholder systems. Congratulations to all the authors.

Reviewer 2 ·

Basic reporting

The article has been improved following the revisions made by the authors following the first round of review. The authors added some useful references to literature about Phaseolous vulgaris pollination and clarified differences between P vulgaris and V faba literature. There are just a few very small changes required.

Raw data does not appear to have been shared - is this a requirement of peerj?
The figure legends state that the error bars are both SE and LSD - please change the legends and clarify which is reported.
Figure 2 y axis label - is this proportion, or is it the number of insects?

Experimental design

Methods around how the yield in kg/ha was calculated are still not clear. This is important because this is the result that the authors report in the abstract. It doesn't appear that the authors measured the total yield mass per plant, only on 30 representative seeds from each plot. I imagine that the authors calculated an average mass per seed from these 30 representative ones, multiplied this by average seeds per pod, then multiplied this by average pods per plant, then by plants per ha - it would be would be useful if the authors included the actual calculation formula used in an appendix.
The authors confirmed in response to reviewers that the field sites were the same for the observation transects and the bagging experiments, but this needs to be written in the paper.
The number of replicates is a bit confusing still, please could the authors use consistent terminology (I think the main to clarify, is whether a plot is the same as a site? If so it would be useful to use only one of these terms). Similarly "four bean plants grown in a block size of 4 m2 and there were four replications per treatment" - are these 4 plants the 4 replicates, or are there 4 replicates of 4 plants (16 per block of each treatment)? If the authors could state the total number of manipulated plants across the whole study within the paper methods section, this would reduce the ambiguity.
Otherwise the methodological details are clearly written and the research is reproducible.

Validity of the findings

No additional comment but see section 2 comments as this applies here too.

Additional comments

The paper is improved now and I enjoyed reading the revised version, thank you.

I spotted a couple of typos -
337 recorded not recoded
352 polliantors not pollinators

357-358 - just to clarify, why would presence of long tongued carpenter bees increase the presence of honeybees? As you say, is the short tongued bee species that typically rob first (e.g. B terrestris) and facilitate secondary robbing by honeybees.

---

## Round 0.3 · accepted · Accept

Thank you for addressing all the pending comments.